# Marine-Derived Components: Can They Be a Potential Therapeutic Approach to Parkinson’s Disease?

**DOI:** 10.3390/md21080451

**Published:** 2023-08-16

**Authors:** Joana Silva, Celso Alves, Francisca Soledade, Alice Martins, Susete Pinteus, Helena Gaspar, Amparo Alfonso, Rui Pedrosa

**Affiliations:** 1MARE—Marine and Environmental Sciences Centre, ARNET—Aquatic Research Network, Polytechnic of Leiria, 2520-630 Peniche, Portugal; franciscasoledade97@gmail.com (F.S.); alice.martins@ipleiria.pt (A.M.); susete.pinteus@ipleiria.pt (S.P.); hmgaspar@ciencias.ulisboa.pt (H.G.); 2MARE—Marine and Environmental Sciences Centre, ARNET—Aquatic Research Network, ESTM, Polytechnic of Leiria, 2520-614 Peniche, Portugal; celso.alves@ipleiria.pt; 3BioISI—Biosystems and Integrative Sciences Institute, Faculty of Sciences, University of Lisbon, 1749-016 Lisboa, Portugal; 4Department of Pharmacology, Faculty of Veterinary, University of Santiago de Compostela, 27002 Lugo, Spain; amparo.alfonso@usc.es

**Keywords:** neurodegenerative disease, ageing society, neuroinflammation, seaweeds, neuroprotective activity, marine bioactive compounds, neurological disorders

## Abstract

The increase in the life expectancy average has led to a growing elderly population, thus leading to a prevalence of neurodegenerative disorders, such as Parkinson’s disease (PD). PD is the second most common neurodegenerative disorder and is characterized by a progressive degeneration of the dopaminergic neurons in the substantia *nigra pars compacta* (SNpc). The marine environment has proven to be a source of unique and diverse chemical structures with great therapeutic potential to be used in the treatment of several pathologies, including neurodegenerative impairments. This review is focused on compounds isolated from marine organisms with neuroprotective activities on in vitro and in vivo models based on their chemical structures, taxonomy, neuroprotective effects, and their possible mechanism of action in PD. About 60 compounds isolated from marine bacteria, fungi, mollusk, sea cucumber, seaweed, soft coral, sponge, and starfish with neuroprotective potential on PD therapy are reported. Peptides, alkaloids, quinones, terpenes, polysaccharides, polyphenols, lipids, pigments, and mycotoxins were isolated from those marine organisms. They can act in several PD hallmarks, reducing oxidative stress, preventing mitochondrial dysfunction, α-synuclein aggregation, and blocking inflammatory pathways through the inhibition translocation of NF-kB factor, reduction of human tumor necrosis factor α (TNF-α), and interleukin-6 (IL-6). This review gathers the marine natural products that have shown pharmacological activities acting on targets belonging to different intracellular signaling pathways related to PD development, which should be considered for future pre-clinical studies.

## 1. Challenges of an Ageing Society: Why Is Advancing Age a Major Risk Factor in Parkinson’s Disease Development?

Demographic trends are a key factor all over the world. The number and proportion of older people in the global population are growing mainly due to the birth rates decreasing and the increase in life expectancy [1]. According to the WHO, this increase will tend to continue throughout this century, as it is expected that by 2050 the number of older people (>60 years) will reach the number of young people (<15 years) [2]. Therefore, population aging nowadays represents one of the most worrying demographic phenomena in modern societies. According to “World Statistics”, the European population aged over 65 in 2020 was about 15.5% of the global population, a significant increase since 1960 (Figure 1). On the other hand, in 2020, the Asian and African continents presented a lower percentage of the aged population over 65, around 5.9% and 3.6%, respectively. However, in those continents the percentage of the population over 65 increased between 1960 and 2020, being more evident in Asia. Life expectancy variation between continents is also influenced by healthcare-related features, such as a lack of proper and widespread recognition of the disorder, difficulties in accessing healthcare systems, and genetic, environmental, and methodological differences. Based on the aging scenario of the world population, especially in Europe, the elderly are the most vulnerable to developing neurodegenerative diseases, as ageing is a critical risk factor. 

Parkinson’s disease prevalence is about 2–3% in the world population aging over 65 years old and 4% in the population over 80s, and rare exceptions may occur in young people between 20 and 50 years [3,4,5]. However, with advancing age, several essential processes related to the neurons’ functions in the substantia *nigra pars compacta*, including dopamine metabolism, the number of mitochondrial DNA (mtDNA) copies, and the decline of protein degradation, can be altered. For instance, dopamine metabolism generates a significant amount of reactive oxygen species that can affect different cellular processes in the neurons. On the other hand, a decline in mtDNA copy number can lead to a decrease in ATP production and a reduction in the protein degradation affecting the functioning of neurons [6]. The accumulation of melanin, the ability of neurons and mitochondria to cope with calcium, and the levels of iron within the cells will be also affected so that additional insults, such as mitochondrial Complex I and IV deficiencies and α-synuclein aggregation, cause the loss of vulnerable neurons, leading to the occurrence of symptoms associated to PD [5]. As age advances, the accumulation of several cellular defects become SNpc neurons more vulnerable. Therefore, the effects of aging can contribute to triggering a cascade of stressor events within the substantia *nigra pars compacta*, weakening the neurons’ response capacity to new insults that are seen as part of the disease process [7]. 

## 2. Mechanisms Underlying Parkinson’s Disease Development

Under normal conditions, the dopaminergic neurons produce the dopamine (DA) neurotransmitter to the striatum responsible to coordinate the functionality of the body muscles and movements through signal transmission [8]. In the absence of dopamine (<80%), the receptors are not stimulated [9], leading to a disruption in the basal ganglia circuits [10,11]. It is estimated that at the onset of PD symptoms, approximately 50–60% of dopaminergic neurons in the SNpc are lost, and at the same time, there is a 70–80% loss of dopamine in the striatum [12]. 

Previous studies have reported that dopaminergic cell death in PD is related to high levels of apoptosis, and several pieces of evidence suggest that oxidative stress, mitochondrial dysfunction, and neuroinflammation events are also involved (Figure 2) [7,8]. 

The brain is an organ particularly susceptible to oxidative damage essentially due to its high metabolic activity and consequent high rate of O_2_ consumption promoting a basal production of ROS, high lipid content, and the relative scarcity of antioxidant enzymes compared to other organs [13,14,15]. Due to those factors, an imbalance between the production of ROS and antioxidant defenses may result in the development of an oxidative stress condition that can play a critical role in mitochondrial dysfunction since the inhibition of mitochondrial Complex I of dopaminergic neuronal cells can lead to a decrease in ATP levels and thus to cell death [16]. The decline of ATP production contributes to the damage of all cellular mechanisms dependent on ATP, also promoting the formation of free radicals leading to a development of oxidative stress conditions [17,18,19,20]. The occurrence of these events promotes mitochondrial dysfunction that is characterized by changes in mitochondrial membrane potential and insufficient production of ATP [21,22]. Several circumstances may contribute to an increase of oxidative stress condition and mitochondrial alterations triggering apoptosis in the central nervous system (CNS), including (a) an increase of DA levels; (b) an increase of iron deposit; (c) glutathione (GSH) deficiency, causing a decrease of brain capacity to eliminate H_2_O_2_; (d) DNA damage; and (e) lipid peroxidation [23,24,25]. According to *post mortem* studies with brains of PD patients, all the neurons in the SNpc seemed to be under a high oxidative stress state, presenting high levels of iron and a decrease in GSH levels and oxidative damage in lipids, proteins, and DNA [22,25]. Thus, due to the critical role of oxidative stress in the neuronal cell death in PD, the antioxidant enzymes (catalase, superoxide dismutase, glutathione peroxidase) that represent the main defense system against oxidative lesions activated by the transcription of Nrf2 factor have been seen as one of the pharmacological strategies to attenuate the oxidative damages induced by ROS [26]. On the other hand, previous studies carried out in PD also demonstrated the existence of a deficiency in Complex I of the respiratory chain limited to the substantia *nigra pars compacta.* These studies evidenced the role of mitochondria in the etiology and pathology of this disease as well as the relationship between oxidative phosphorylation and free radical production [27,28,29], pointing out the role of mitochondria as the main source of ROS that contribute to the development of an intracellular oxidative stress condition [30]. 

In the last few years, several studies have reported the involvement of neuroinflammatory mechanisms that trigger a cascade of events leading to the degeneration of neuronal cells [31]. Chronic neuroinflammation is one of the hallmarks of PD pathophysiology and, under inflammatory conditions, activated glial cells release pro-inflammatory factors, such as cytokines and prostaglandins, as well as neurotoxic factors, such as reactive oxygen species (ROS)/reactive nitrogen species (RNS) and complement proteins that can exacerbate the neuronal cell damage, leading to neurodegeneration [17,32]. 

Although an oxidative stress condition and mitochondrial dysfunction are closely related to the death of dopaminergic neurons, in the last few years, several studies have reported the involvement of neuroinflammatory mechanisms that trigger a cascade of events also leading to the degeneration of neuronal cells. The first evidence of inflammation in PD was reported by McGerr and collaborators (1988) [33] that observed activated cGMP in *post mortem* PD patients mainly in the SNpc and in the putamen, two of the areas more affected in this disease. In fact, neuroinflammation that is primarily controlled by microglia seems to play a critical role in the neuronal cell loss observed in PD [11,34]. Microglia are immunocompetent cells in the CNS that are responsible for protecting the brain against invading pathogens and capable of stimulating an immune-adaptive response [35,36]. Therefore, several acute lesions that may occur in the CNS led to microglial activation, triggering a series of microglia alterations, including the increase in proliferation and the production of inflammatory mediators [37]. However, this inflammatory process can be considered beneficial for the neuronal tissue since it promotes the elimination of cellular debris and the secretion of neurotrophic factors. During the PD neuroinflammation processes, microglia release pro-inflammatory cytokines, such as TNF-α, IL-1β, and IL-6. These cells act on the endothelium of the blood–brain barrier (BBB), increasing its permeability and stimulating the expression of adhesion molecules and chemokines that recruit peripheral blood mononuclear cells, such as macrophages, monocytes, and T lymphocytes contributing to the development of a neuroinflammatory condition through the production and secretion of the above described pro-inflammatory mediators [38,39].

All these unfavorable events play an important role in PD, contributing to dopaminergic neurodegeneration. Thus, the development of new therapeutic strategies with the purpose of preventing or delaying the neurodegeneration observed in PD is of utmost importance.

## 3. Marine Natural Products as New Potential Therapeutic Agents for Parkinson’s Disease 

From a very early age, humans began to use the oceans to navigate and search for food and new territories, but it was only in the middle of the 19th century that they began to explore the seabed. In the last few decades, the marine environment has proven to be a unique source of new chemical entities with potential applications in different areas, such as pharmaceutical, cosmetics, food, and agrochemical, revealing itself as a vital factor for the discovery of new products [40].

The number of marine species that inhabit the world’s oceans is not truly known; however, experts estimate a number that approaches one to two million species [41]. These organisms survive and live in a competitive and exigent environment constituted by complex communities, establishing close associations with other organisms. These interactions impose several ecological challenges, such as competition for space, predation, and tidal variations, among others, leading marine organisms to develop distinct strategies, including survival and behavioral, physical, and/or chemical [42,43]. In fact, along their evolution, marine organisms developed the capacity to produce secondary metabolites with unusual chemical structures that act as chemical weapons against those ecological threats.

Over the past few decades, several compounds have been isolated from marine organisms, and many of them have exhibited great biological activities, including anticancer, antimicrobial, antifungal, antiviral, anti-inflammatory, etc. [43,44,45], inspiring the development of new therapeutic agents [46,47]. According to Faulkner and Blunt and their collaborators, between 1977 and 2019, sponges (30.93%), followed by microorganisms (20.53%) and seaweeds (10.44%), were the marine organisms from which the largest number of new natural compounds were isolated (Figure 3).

Due to the urgent need to discover and to develop new effective therapeutic agents for neurodegenerative diseases treatment, including PD, as well as the ability of marine organisms to biosynthesize compounds with distinct chemical structures, biological activities, and mechanisms of action, several scientists have explored their ability as neuroprotective agents [10]. By analyzing the compounds with neuroprotective activities isolated from marine organisms (Table 1), it is possible to observe that most of them were isolated from sponges, fungi, seaweeds, and corals, followed by starfish, cnidarians, and bacteria. 

The chemical structures (Figure 4, Figure 5, Figure 6, Figure 7, Figure 8, Figure 9, Figure 10 and Figure 11) and mechanisms of action of the marine natural products presented in Table 1 are deeply described in Section 3.1, Section 3.2, Section 3.3, Section 3.4, Section 3.5, Section 3.6, Section 3.7, Section 3.8 and Section 3.9.

### 3.1. Bacteria

The neuroprotective activity of marine compounds derived from bacteria is still poorly explored [3,4,126,127]. NP7 (**1**) (Figure 4) is a marine-derived compound from *Streptomyces* sp. that prevents apoptosis and necrosis induced by H_2_O_2_ treatment on neurons and glial cells. Furthermore, NP7 also displayed the ability to inhibit microglial activation and prevented the increase in ERK phosphorylation induced by H_2_O_2_ exposure [67]. Mannosylglycerate (**2**) (Figure 4) isolated by Faria and coworkers (2013) [70] from aggregation/expression thermophilic bacteria reduced the number of *Saccharomyces cerevisiae* cells that presented α-synuclein as well as inhibited the fibrillation of this protein. These results are relevant since α-synuclein is the main component of the intraneuronal inclusions found in the brains of PD patients. Piloquinone A (**3**) and piloquinone B (**4**) (Figure 4) obtained from *Streptomyces* sp. have been shown to be potent inhibitors of MAO-A (IC_50_ = 6.47 µM) and MAO-B (IC_50_ = 1.21 µM), suggesting their therapeutic potential in PD as MAO inhibitors.

### 3.2. Fungi 

Neoechinulin A (**5**) (Figure 5), an alkaloid isolated from different species of marine fungi, demonstrated the ability to decrease the neurotoxic effects induced by 1-methyl-4-phenylpyridinium (MPP^+^) toxin and rotenone on PC12 cells. Kajimura et al. (2008) [72] observed that neoechinulin A could protect PC12 cells from MPP^+^-induced neurotoxicity mediated by inhibition of mitochondrial dysfunction in Complex I, suggesting that this alkaloid can ameliorate downstream events of mitochondrial failure. Furthermore, Akashi et al., (2011) [73] also observed that neoechinulin A could protect PC12 cells from rotenone-induced neurotoxicity. The authors verified that pre-treatment with neoechinulin A significantly impeded cell death and decreased ATP levels promoted by rotenone. Lu and coworkers (2010) [74] isolated xyloketal B (**6**) (Figure 5) from the fungus *Xylaria* sp., which decreased the neurotoxic effects induced by MMP^+^ in vitro and in vivo models, namely PC12 cells and *Caenorhabditis elegans*, respectively. Xyloketal A attenuated the MMP^+^ toxin effects through the decrease in the intracellular ROS levels and restoration of the GSH levels. Secalonic acid A (**7**) (Figure 5), obtained from the fungus *Aspergillus ochraceus*, inhibited the neurotoxicity induced by the MMP^+^ toxin on rat cortical neurons. The compound promoted the activation of c-jun N-terminal Kinases (JNK) and MAPK-p38 and the inhibition of Caspase-3 activity, decreasing apoptosis triggered by MPP⁺ [76]. Compound 6-hydroxy-*N*-acetyl-β-oxotryptamine (**8**) (Figure 5), isolated from *Penicillium* sp., protected Neuro2A cells against the damage induced by 6-OHDA and Paraquat (PQ) neurotoxins, mediating a reduction in ROS production more marked than the antioxidant standard melatonin. Finally, the 3-methylorsellinic acid (**9**) and 8-methoxy-3,5-dimethylisochroman-6-ol (**10**) compounds isolated from *Penicillium* sp., candidusin A (**11**), and 4”-dehydroxycandidusin (**12**) attained from *Aspergillus* sp. and the compound diketopiperazine mactanamide (**13**) (Figure 5) isolated from *Aspergillus flocculosus* also exhibited protective effects against 6-OHDA and PQ neurotoxin treatments on Neuro2A cells, decreasing the production of ROS. Asperpendoline (**14**), a prenylated indole alkaloid with a diketopiperazine motif, was discovered from the co-cultivation of *Aspergillus ochraceus* and *Penicillium* sp. This compound, which presents a rare skeleton of pirymido[1,6-a] indole, was screened for neuroprotective activity through its administration to H_2_O_2_-injured SH-SY5Y cells. Upon administration, cells displayed attenuated ROS accumulation and enhancement of GSH levels. 

### 3.3. Mollusks

Docosahexaenoic acid-(**15**) and eicosapentaenoic acid (**16**)-enriched phospholipids (DHA/EPA-PLs) (Figure 6) [81] supplemented and isolated from mollusks (*Sthenoteuthis oualaniensis*) exhibited the ability to protect against MPTP-induced impairments in PD mice. The treatment with DHA/EPA-PL has the ability to recover brain DHA levels and exert neuroprotective effects in old age in long-term n-3 PUFA-deficient mice, improving the loss of dopaminergic neurons, which induced mitochondrial dysfunction and the neurodevelopment delay. Wang et al. (2016) [80] studied the neuroprotective effect of different n-3 PUFA formulations isolated from the *Sthenoteuthis oualaniensis* mollusk against brain oxidative injury induced by MPTP in male C57BL/6J mice. The mechanism underlying these effects was evaluated by the authors, showing the pre-treatment with DHA/EPA-PL (**15**,**16**) more effectively suppressed MPTP-induced apoptosis by influencing the Bax/Bcl2 ratio and Caspase-9, Caspase-3, and cytoplasmic cytochrome C levels. Additionally, supplementation of DHA/EPA-PL also showed reduced phosphorylation of p38 and Jun N-terminal kinase (JNK). The results obtained indicated that DHA/EPA-PL could offer an efficient strategy to explore novel functional foods, providing neuroprotective potential. Docosahexaenoic acid (DHA) (**15**) [79] enriched phospholipids also isolated from *Sthenoteuthis oualaniensis* were investigated regarding the effects of DHA-enriched phospholipids with different polar groups (DHA-PC and DHA-PS) on MPTP-induced PD mice, increasing the number of dopaminergic neurons and inhibiting apoptosis through the mitochondrial pathway and MAPK pathway. The treatment of DHA with different polar groups exerted varying improvements in PD mice, which represented a potential novel therapeutic candidate for the prevention and treatment of neurodegenerative diseases. Astaxanthin is a natural pigment that has been shown to be an excellent antioxidant and contender for testing neurological diseases. This compound (**17**) [82] (Figure 6) was isolated from a mollusk, *Doryteuthis singhalensis*, and the neuroprotective activity was evaluated by preventing rotenone-induced toxicity of SK-N-SH human neuroblastoma cells in an in vitro model of experimental Parkinsonism. Anguchamy et al. (2023) [82] revealed that the cells exposed to 100 nM rotenone for 24 h exhibited 52% of cell viability, and when pre-treated with various concentrations of astaxanthin for 2 h, cell viability was significantly increased with higher doses: 15, 20, 25, and 30 nM. Additionally, the astaxanthin treatment showed to reduce mitochondrial membrane potential and ROS production during rotenone administration, suggesting that astaxanthin protects SK-N-SH cells from rotenone toxicity. The authors also tested the level of TBARS and GSH in rotenone-treated SK-N-SH cells incubated with or without astaxanthin and revealed that pre-treatment with astaxanthin (25 nM) significantly decreased the levels of TBARS, improving the antioxidant defense system of rotenone-induced cells.

### 3.4. Sea Snails

α-Conotoxin (**18**) (Figure 7), isolated from the sea snail *Conus textile* [84], exhibited the ability to regulate DA release blocking Nicotinic acetylcholine receptor α6/α3β2β3 nAChR expression on a rat nAChRs model, showing an IC_50_ value of 28 mM. It is a promising candidate for new PD therapeutics. Manigandan et al. (2019) [85] studied the neuroprotective effects of asulfated chitosan isolated from *Sepia pharaonis* on SH-SY5Y cells against the toxicity induced by rotenone. The sulfated chitosan pretreatment increased SH-SY5 cell viability by attenuating the ROS production levels, increasing antioxidant enzymes and inhibiting mitochondrial dysfunction and apoptosis. YIAEDAER, a multi-functional peptide, was isolated and purified from the visceral mass extract of *Neptunea arthritica cumingii*. When YIAEDAER was administered to MPTP-induced zebrafish, its locomotor impairment was suppressed, the degeneration of DA neurons was ameliorated, and the loss of cerebral vessels was inhibited. 

### 3.5. Sea Cucumber 

Glucocerebrosides, SCG-1 (**19**), SCG-2 (**20**), and SCG-3 (**21**) (Figure 8) isolated from *Cucumaria frondosa* are important sphingolipids. Wang et al. (2018) [87] studied in a nerve growth factor (NGF)-induced PC12 model the neuritogenic effect and signaling pathways of SCGs. Neuritogenesis is a dynamic phenomenon associated with neuronal differentiation, and it is important for neuronal development and regeneration. Neurotrophic factors including NGF, brain-derived neurotrophic factor (BDNF), and neurotrophins 3 and 4/5 (NT3 and NT4/5, respectively) can stimulate neurite growth, maintain viability, and induce differentiation. This way, the authors observed that SCGs exhibited significant neuritogenic effects in a dose-dependent and structure-selective manner, which increased the ratio of neurite-bearing cells and expression of axonal (GAP-43) and synaptic (synaptophysin) proteins. In addition, mechanistic studies suggested that SCGs reinforced the NGF-induced phosphorylation of TrkA, coupled with ERK1/2 activation, triggering the activation of cAMP response element-binding protein (CREB) that results in a marked increase in BDNF expression. 

Decanoic acid (**22**) was isolated from ethyl acetate fractions of the black sea cucumber *H. leucospilota* and was administered to two models of *C. elegans*, a 6-OHDA-induced Daergic neurodegeneration worm model and *C. elegans* expressing alfa-synuclein model. After administration with decanoic acid, it was possible to observe an attenuation of Daergic neurodegeneration, reduction in alfa-synuclein aggregation, and improvement of the behavioral deficits associated with PD. Moreover, decanoic acid was able to activate the DAF-16 transcription factor, leading to the upregulation of its targeted genes, including antioxidant genes and small heat shock proteins, thus enhancing oxidative stress and proteotoxic stress resistance. 

Palmitic acid (**23**), also isolated from ethyl acetate fractions of the black sea cucumber *H. leucospilota*, was tested for neuroprotection in a *C. elegans* PD model. It was observed that when administered to this model, palmitic acid was able to ameliorate the loss of Daergic neurons, improve dopamine-dependent behaviors, and rescue the lifespan of worms. Moreover, when tested in *C. elegans*-expressing alfa-synuclein, worms displayed a decrease in alfa-synuclein aggregation, improvement of the motor deficits associated with PD, and a prolonged lifespan.

Eicosapentaenoic acid enriched phospholipids (EPA-PL) (16) [92], isolated from the sea cucumber *Cucumaria frondosa*, were tested for neuroprotection on PD mice induced by MPTP. It was observed that when administered to this model, Eicosapentaenoic acid was able to improve MPTP-induced behavioral deficiency. Further research showed that EPA-PL suppressed MPTP-induced oxidative stress and apoptosis, thereby alleviating the loss of dopaminergic neurons via the mitochondria-mediated pathway and mitogen-activated protein kinase pathway, providing a reference for the development of functional ingredients and dietary guidance to prevent neurodegenerative diseases.

### 3.6. Seaweeds

Seaweeds have been revealed to be a huge source of new chemical entities with great biological effects, and they have been widely used for nutritional or medicinal purposes for many years [128], but their potential source of compounds with neuroprotective activities is still underexplored. Jin et al. (2014) [93] studied the neuroprotective effects of six heteropolysaccharides isolated from the genus *Sargassum*, namely *Sargassum integerrimum*, *Sargassum nozhouense,* and *Sargassum fusiformis*. On the other hand, Wang and coworkers (2016) isolated the porphyran polysaccharide from the seaweed *Porphyra haitanensis* together with two of its derivatives [99]. All these compounds inhibited the neurotoxicity induced by 6-OHDA on MES23.5 dopaminergic neuron cells. Fucoidans are sulfated polysaccharides mainly present in brown seaweeds, and their neuroprotective activity was already evaluated on different cellular and in vivo models [72,73,75,76,82]. Jin and coworkers (2013) [94] described the neuroprotective potential of fucoidan isolated from *Saccharina japonica* on MES23.5 and SH-SY5Y cells against the toxicity induced by 6-OHDA neurotoxin. On the other hand, Gao and coworkers (2012) [96] evaluated the neuroprotective effect of fucoidan isolated from the brown seaweed *Laminaria japonica* on PC12 cells when exposed to H_2_O_2_. The polysaccharide treatment increased PC12 cells’ viability, attenuating the H_2_O_2_-induced toxicity and reducing the ROS generation and lactate dehydrogenase (LDH) release [96]. Fucoidan also inhibited apoptosis, increasing the Bcl-2/Bax ratio and decreasing the expression of Caspase-3 as well as enhancing the PI3K/Akt signaling pathway. Furthermore, in vivo assays were also conducted with fucoidan. Meenakshi and coworkers (2016) [97] reported the neuroprotective effects of fucoidan on rats, observing that in the presence of fucoidan, an increase in antioxidant defenses and DA levels occurred when compared to MPTP treatment. Moreover, the authors also observed an increase in tyrosine hydroxylase (TH) expression in the fucoidan-treated group, which correlates with the levels of TH protein in the substantia nigra and corpus striatum. In addition, it was verified that the increase was greater than the content of dopamine and 3,4-dihydroxyphenylacetic acid (DOPAC), which may explain that the dopaminergic terminals are more sensitive to MPTP toxicity and, therefore, are more severely damaged than the dopaminergic cell bodies. Huang et al. (2018) [100] also reported the neuroprotective effects of fucoidan isolated from the brown seaweed *Sargassum hemiphyllum* against the toxicity induced by 6-OHDA neurotoxin on SH-SY5Y cells. Moreover, the authors observed a decrease in cytochrome *C* release, Caspase-3, -8, and -9 activities, and protection against DNA fragmentation and phosphorylation of Akt. Fucoidan isolated from the brown seaweed *Fucus vesiculosus* demonstrated a neuroprotective effect both in vitro and in vivo. When fucoidan was administered to an MPP+-induced SH-SY5Y cell model, cellular viability was enhanced and morphological damage was attenuated. Administration of fucoidan to an MPTP-induced PD mouse model (C57BL/6 mice) alleviated motor deficits and slowed the progression of dopamine neuron loss, protecting the function of the nigral striatum.

Liu and coworkers (2022) [129] reported the neuroprotective effects of fucoxanthin (**24**) (Figure 8) isolated from brown seaweeds on the 6-OHDA-lesioned mice model of PD. The authors observed that long-term administration of L-dopa (30 mg/kg) produced neurotoxicity through Caspase-3 and c-Jun expression mediated by the ERK/JNK system; however, with fucoxanthin treatment, a decrease in ERK1/2, JNK1/2, c-Jun phosphorylation, and Caspase-3 was verified, improving the exercise ability of PD mice. Phlorotannins, a group of phenolic compounds found in brown seaweeds, exhibit a variety of biological properties, including antioxidant, neuroprotective, and memory-enhancing effects, among others. Accordingly, Cha and coworkers (2016) [102] evaluated the neuroprotective effects of dieckol (**25**) (Figure 9), a phlorotannin isolated from *Ecklonia cava*, on rotenone-induced oxidative stress in SH-SY5Y cells. The authors observed a reduction in intracellular ROS production, cytochrome *C* release, and mitochondrial function protection and a retardation of α-synuclein aggregation by treatment with rotenone. Du and co-workers (2021) [104] also reported the neuroprotective effects of two alginate oligosaccharides prepared from enzymatic degradation, polymannosic acid (**26**), and polyguluronic acid (**27**) (Figure 9) isolated from brown seaweed on a mice model of PD. The authors observed that, in the presence of these two acids, only polymannosic acid could improve the motor functions of PD mice. However, they verified that polymannosic and polyguluronic acids can prevent dopaminergic neuronal loss by increasing TH expression in the midbrain of PD mice. On the other hand, polyguluronic acid could inhibit inflammation through an increase in the striatal neurotransmitter 5-hydroxyindole acetic acid levels. Souza and coworkers (2017) [108] demonstrated the neuroprotective effect of agaran, a polysaccharide isolated from the seaweed *Gracilaria cornea*, on rats subjected to the 6-OHDA neurotoxin. The authors observed that agaran promoted neuroprotective effects in vivo by reducing oxidative/nitroactive stress and through the alteration in the MAO contents induced by 6-OHDA.

The occurrence of an inflammatory response in the central nervous system has been associated with the development of neurodegenerative diseases, such as PD, which is characterized by the activation of microglial cells and an increase in inflammatory cytokines expression. Accordingly, Yao et al. (2014) [106] evaluated the neuroprotective effects of κ-carrageenan isolated from a red seaweed, which inhibited the activation of microglial cells and decreased TNF-α and arginase expression after treatment with LPS. In turn, κ-carrageenan attained from the red seaweed *Hypnea musciformis* [107] protected SH-SY5Y cells against the neurotoxicity induced by 6-OHDA, reducing H_2_O_2_ production, Caspase-3 activity, and MMP depolarization. Souza and coworkers (2017) [108] demonstrated that the sulfated polysaccharide isolated from the red seaweed *Gracilaria cornea* reverted the damage induced by 6-OHDA treatment changing monoamine content and reducing oxidative stress condition. On the other hand, Silva and coworkers (2021) [105] evaluated the neuroprotective effect of two compounds, the linear diterpene eleganolone (**28**) and the monoterpene lactone loliolide (**29**) (Figure 9) isolated from the brown seaweed *Bifurcaria bifurcata* and from the green seaweed *Codium tomentosum*, respectively, on SH-SY5Y cells treated with 6-OHDA. Treatment with eleganolone increased SH-SY5Y cell’s viability, reducing 6-OHDA-induced neurotoxicity, reducing ROS generation, H_2_O_2_ production, and Caspase-3 activity, increasing Catalase activity and ATP levels, and inhibiting the translocation of NF-kB factor. On the other hand, loliolide treatment also increased SH-SY5Y cell’s viability, reducing 6-OHDA-induced neurotoxicity, ROS generation, and Caspase-3 activity, protecting mitochondrial membrane potential, and inhibiting the DNA fragmentation and translocation of NF-kB factor.

### 3.7. Soft Coral 

Austrasulfone (**30**), 1-tosylpentan-3-one (**31**), and 11-dehydrosinulariolide (**32**) compounds (Figure 10) isolated from soft corals have been shown to protect SH-SY5Y cells from neurotoxicity induced by 6-OHDA acting in different intracellular signaling pathways. The metabolite 1-tosylpentan-3-one (**31**) (Figure 9) isolated by Kao et al. (2017) [113] from *Cladiella australis* exhibited neuroprotective effects in SH-SY5Y cells treated with 6-OHDA mediated by MAPK-p38 activation, inhibition of Caspase-3 activity, and increased expression of factor Nrf2 and oxygenase-1 (HO-1) through the intracellular signaling pathway PI3/AKT. In addition, the authors also tested the neuroprotective effects of 1-tosylpentan-3 on a zebrafish model, reporting an improvement in locomotive capacity and a decrease in TNF-α interleukin expression (Table 1). Chen and collaborators (2012) [114] isolated 11-dehydrosinulariolide (**32**) (Figure 10) from *Sinularia flexibilis*, which exhibited anti-inflammatory and neuroprotective activities. Regarding the anti-inflammatory effect, 11-dehydrosinulariolide reduced the expression of two major pro-inflammatory mediators, NO and cyclooxygenase, on lipopolysaccharide-stimulated macrophages. On the other hand, the neuroprotective activity was evaluated on SH-SY5Y cells exposed to 6-OHDA neurotoxin. The compound exhibited the ability to recover the 6-OHDA-induced neurotoxicity on SH-SY5Y cells. This protective effect was accompanied by a significant inhibition of Caspase-3/7 activities as well as by the mediation of PI3K signaling, activating phosphorylated-AKT. Feng and coworkers (2016) [130] also evaluated the neuroprotective effects of 11-dehydrosinulariolide on different in vitro and in vivo models exposed to 6-OHDA neurotoxin, namely SH-SY5Y cells, rats, and zebrafish. The authors observed that the compound enhanced the DJ-1 expression in the SH-SY5Y cells cytoplasm and activated the AKT/ PI3K, Nrf2/HO-1 intracellular signaling pathways. In addition, it also revealed a 6-OHDA-induced attenuation on tyrosine hydroxylase (TH) expression, a dopaminergic neuronal marker, and in zebrafish and rat models of PD, it could also be reversed by treatment with 11-dehydrosinulariolide.

Bradia et al. (1998) [115] isolated a new diterpene sarcophytolide (**33**) (Figure 10) from *Sarcophytom glaucum*, which exhibited cytoprotective effects against glutamate-induced neurotoxicity in primary cortical cells from rat embryos, resulting in significant increase of the percentage of viable cells. The authors observed that the cytoprotective effect presented by sarcophytolide was mediated by a decrease in Ca^2+^ levels and an increase in Bcl-2 pro-oncogene expression levels. 

### 3.8. Sponge 

The gracilins A, H, K, J, L (**34**–**38**) and tetrahydroaplysulphurin -1 (**39**) (Figure 11) are a group of diterpenes previously isolated from *Spongionella gracilis* that exhibited the capacity to act on tyrosine kinases. Moreover, all compounds demonstrated the ability to protect cortical neurons against oxidative damage induced by H_2_O_2_ mediated by restoring mitochondrial membrane potential, scavenging intracellular ROS, increasing glutathione levels, decreasing Caspase-3 expression, and activating the Nrf2/ARE pathway [131]. On the other hand, the alkaloid sarain A (**40**) and two bromopyrrole alkaloids, hymenin (**41**) and hymenialdisine (**42**) (Figure 11) isolated from the sponge *Haliclona (Rhizoniera) sarai*, exhibited the same effects previously described [16,117]. Boccito and coworkers (2017) [118] isolated the alkaloid psammaplysene A bromotyrosine (**43**) (Figure 10) from the *Psammaplysilla* sp. sponge that displayed neuroprotective effects in HEK293 cells and in a *Drosophila* in vivo model that seem mediated by modifying processes dependent on heterogeneous nuclear ribonucleoprotein (HNRNPK) that controls biological aspects of RNA [118]. Two sterols, 24-hydroperoxy-24-vinylcholesterol (**44**) and 29-hydroperoxystigmasta-5.24(28)-dien-3-ol (**45**) (Figure 11), isolated from *Xestospongia testudinaria*, also inhibited H_2_O_2_-induced neurotoxicity and activated NF-kB transcription factor, respectively [132]. Iotrochotazine A (**46**) (Figure 10), derived from the *Iotrochota* sp. sponge, was used as a chemical probe in a phenotypic assay panel based on human olfactory neurosphere-derived cells (hONS) from idiopathic PD patients. The authors verified that iotrochotazine A did not exhibit cytotoxicity but affected the morphology and the cellular distribution of lysosomes and endosomes in olfactory neurosphere-derived cells. These results suggest that the compound may be a useful tool to investigate the molecular mechanisms underlying PD [120]. In addition, Wang et al. (2016) isolated jaspamycin (**47**) (Figure 11), an alkaloid from *Jaspis splendens*, which increased the lysosomal staining and decreased the number of EEA-1 characteristic of phenotypic response in a hONS neurosphere-derived cell model of PD [121]. Aerothionin (**48**) and aerophobin-2 (**49**) [122], two bromotyrosine derivatives, were isolated from *Aplysinella* sp. due to their ability to bind to alfa-synuclein and to inhibit its aggregation in a ThT aggregation assay. Because aerothionin has been demonstrated to be toxic to primary dopaminergic neurons, it was removed from further assays. Opposingly, aerophobin-2 was not toxic to primary dopaminergic mouse neurons and, when further assessed, demonstrated the ability to inhibit α-synuclein aggregation in this neuronal population [122]. 

### 3.9. Starfish

From the starfish *Asterias rollestoni*, Zhang and coworkers (2013) [133] isolated two polysaccharides, glucan and mannoglucan sulfate, which were able to inhibit the neurotoxic effects induced by 6-OHDA in mouse embryonic stem cells (MES 23.5 cells).

## 4. Marine-Derived Compounds in Clinical Trials for Parkinson’s Disease

Marine-derived natural compounds have a wide range of pharmacological effects, making them valuable candidates for the development of novel drugs. Five marine-derived natural compounds have already reached the clinical trial stage for PD treatment. These results are presented in Table 2, and their structures are depicted in Figure 12.

The therapeutic potential of docosahexaenoic acid, an omega-3 fatty acid, was evaluated in two clinical trials for PD treatment. The first study started in 2012 and finished in 2016. Twenty-nine PD patients received docosahexaenoic acid supplements along with antidepressants for 12 weeks. In the second trial, which involved sixty PD patients, researchers evaluated the effects of omega-3 fatty acids and vitamin E supplementation. In both trials, a significant relief of depressive symptoms was observed, suggesting the safety and potential use of docosahexaenoic acid for PD treatment. 

From sponges, a precursor of uric acid, the inosine (**51**), which was targeted in a clinical study that started in 2016 and finished in 2019, was isolated. This molecule was effective in raising serum and cerebrospinal fluid urate levels in PD patients, revealing a potential disease-modifying therapy for PD.

Pramipexole (**52**) was isolated from a marine yeast. This compound is a dopamine agonist and can be used in the treatment of PD. A clinical trial study carried out between 2012 and 2017 aimed to study the effectiveness of the compound as a placebo in resolving depressive symptoms in PD patients.

Two clinical trial studies performed between 2002–2005 and 2006–2014 observed that the compound CEP-1347 (**53**), a semisynthetic indolocarbazole derivative isolated from naturally occurring *Nocardiopsis* sp., could inhibit the SAPK/JNK pathway, which is activated after a variety of neuronal toxic insults in neurons. However, this compound did not show effective therapeutic effects in early PD patients. 

The ganglioside GM1 (**54**) isolated from marine bacteria was tested in clinical trial studies between 1999 and 2010, showing that PD patients taking this compound evidenced early improvements and a slow progression of symptoms. However, the detailed mechanism of the neuroprotective effects of GM1 ganglioside is still uncertain.

## 5. Conclusions and Final Remarks

Worldwide, population aging is contributing to the increase in neurodegenerative disorders, including PD. The currently available therapeutics only alleviate the disease symptoms, and the search for new effective drugs to treat and improve the elderly quality of life is urgent. Marine organisms can biosynthesize molecules with unique structural features with great potential for therapeutic applications against different neurodegenerative diseases. In this review, about 60 marine natural products with distinct structures (peptides, alkaloids, quinones, terpenes, polysaccharides, polyphenols, lipids, pigments, and mycotoxins) isolated from diverse marine organisms (bacteria, fungi, mollusk, sea cucumber, seaweeds, soft coral, sponge, and starfish) with neuroprotective potential on PD therapy are summarized. The reported studies showed that those marine natural products have the potential to act on different PD targets, suggesting they can act in several PD hallmarks (reducing oxidative stress, preventing mitochondrial dysfunction, α-synuclein aggregation, and blocking inflammatory pathways through the inhibition of Caspase cascade, DNA fragmentation, the translocation of NF-kB factor, the reduction of TNF-α and IL-6, and the retardation of α-synuclein aggregation). Additionally, some marine organisms also showed a down-regulation of Bax expression, suppression of Caspase-3 activation and inhibition of the phosphorylation of JNK and p38 MAPK, inhibition of MAO, and restoration of total GSH levels. Nevertheless, one of the main bottlenecks associated with the development of new therapeutics for PD is the difficulty of many drugs to reach the brain due to the blood–brain barrier (BBB)—a highly selective semi-permeant barrier that protects the brain and the spinal cord from foreign substances. From the 60 natural marine products reported in this study, only ten compounds exhibited the ability to cross the BBB. Therefore, it is critical to develop more effective therapeutic agents for this neurodegenerative disease with fewer side effects than the currently available drugs and with the capacity to cross the BBB. On the other hand, the development of innovative delivery systems, such as nanoparticles, may contribute to overcoming such limitations of some drugs to cross the BBB. Therefore, there is a window of opportunities for researchers to evaluate the potential of marine-derived compounds as innovative drugs for the treatment of neurodegenerative diseases like PD. 

## Figures and Tables

**Figure 1 marinedrugs-21-00451-f001:**
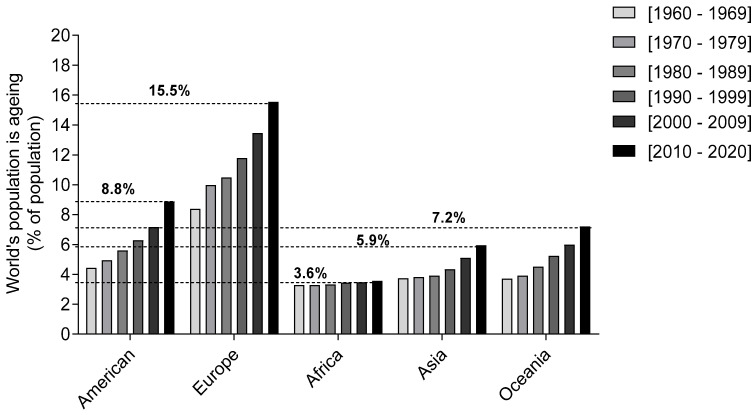
Percentage of global population with age above 65 years old. Adapted from the “World Statistic” data platform, 2021.

**Figure 2 marinedrugs-21-00451-f002:**
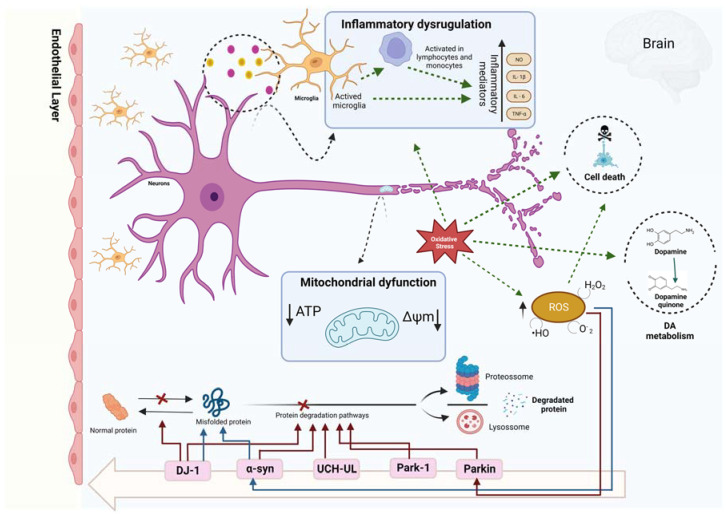
Physiological processes related to Parkinson’s disease pathogenesis. Created with BioRender.com (accessed on 6 July 2023).

**Figure 3 marinedrugs-21-00451-f003:**
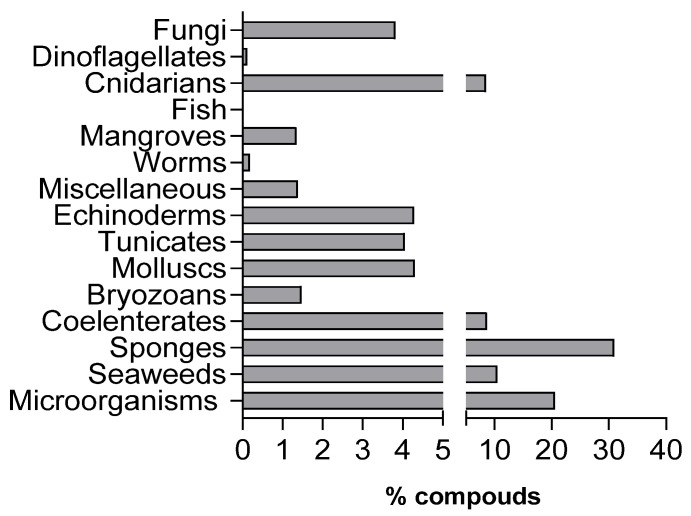
New marine natural products (%) isolated from marine organisms between 1977 and 2021 (Adapted from Faulkner, 1984–2002; Blunt et al., 2003–2018; Carroll et al., 2019; Carroll et al., 2020; Carroll et al., 2021; Carroll et al., 2022; Carroll et al., 2023) [48,49,50,51,52,53,54,55,56,57,58,59,60,61,62,63,64,65,66].

**Figure 4 marinedrugs-21-00451-f004:**
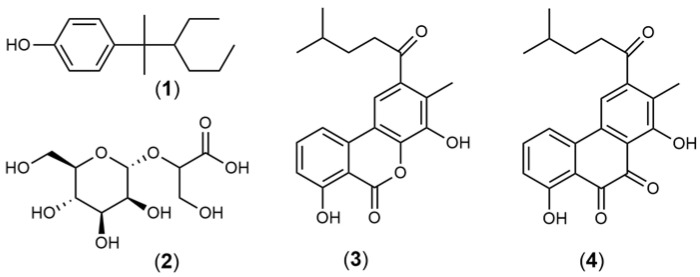
Chemical structures of marine natural products isolated from bacteria: NP7 (**1**), mannosylglycerate (**2**), piloquinone A (**3**), piloquinone B (**4**).

**Figure 5 marinedrugs-21-00451-f005:**
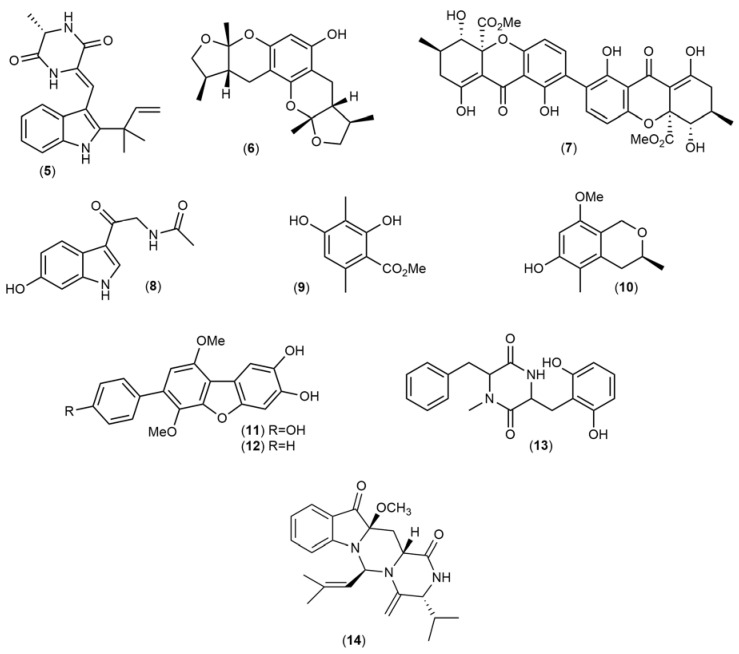
Chemical structures of marine natural products isolated from fungi: neoechinulin A (**5**), xyloketal B (**6**), secalonic acid A (**7**), 6-hydroxy-*N*-acetyl-b-oxotryptamine (**8**), 3-methylorsellinic acid (**9**), 8-methoxy-3,5-dimethylisochroman-6-ol (**10**), candidusin A (**11**), 4``-dehydroxycandidusin A (**12**), diketopiperazine mactanamide (**13**), and asperpendoline (**14**).

**Figure 6 marinedrugs-21-00451-f006:**
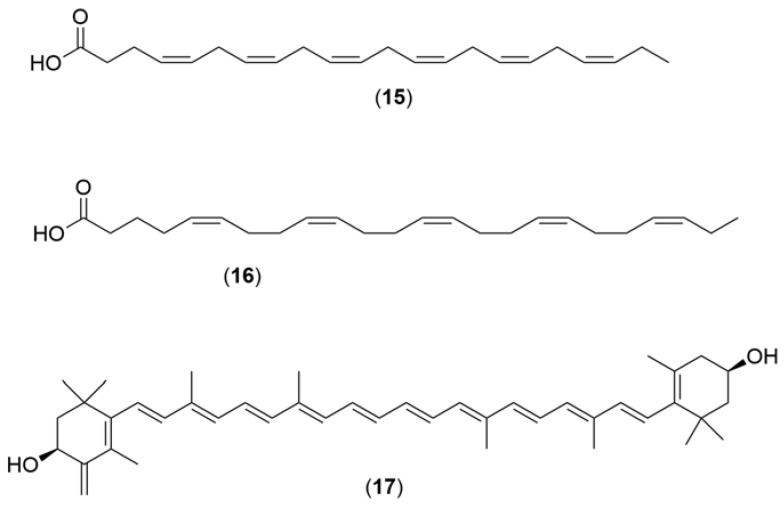
Chemical structures of marine natural products isolated from mollusks: Eicosapentaenoic acid (**15**), docosahexaenoic acid (**16**), and astaxanthin (**17**).

**Figure 7 marinedrugs-21-00451-f007:**
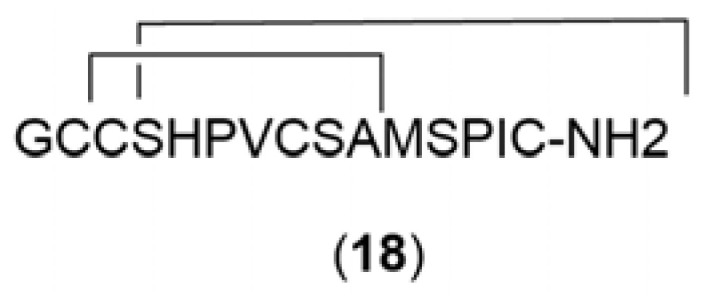
Chemical structures of marine natural products isolated from sea snails: α-Conotoxin (**18**).

**Figure 8 marinedrugs-21-00451-f008:**
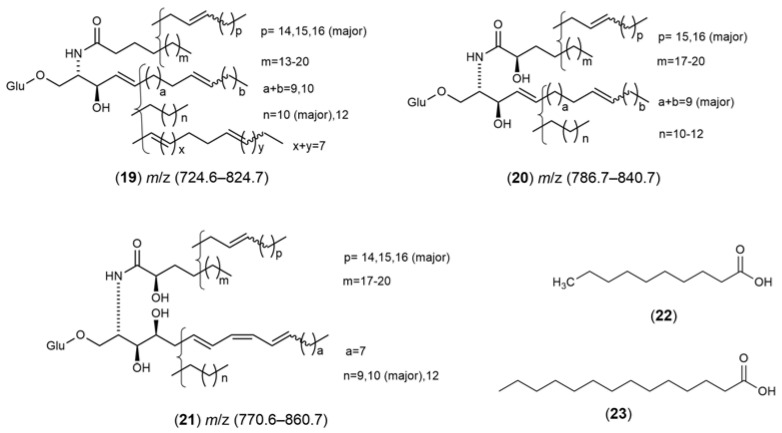
Chemical structures of marine natural products isolated from sea cucumber: SCG-1 (**19**), SCG-2 (**20**), SCG-3 (**21**), decanoic acid (**22**), and palmitic acid (**23**).

**Figure 9 marinedrugs-21-00451-f009:**
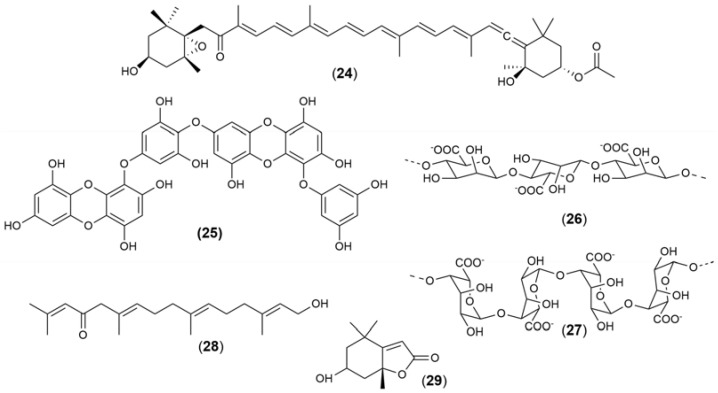
Chemical structures of marine natural products isolated from seaweeds: fucoxanthin (**24**), dieckol (**25**), polymannosic acid (**26**), polyguluronic acid (**27**), eleganolone (**28**), and loliolide (**29**).

**Figure 10 marinedrugs-21-00451-f010:**
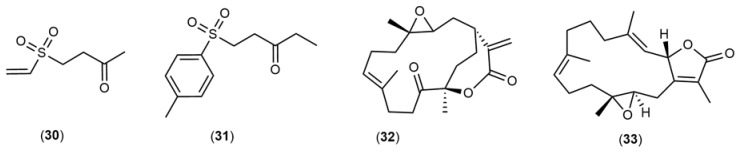
Chemical structures of marine natural products isolated from soft corals: austrasulfone (**30**), 1-tosylpentan-3-one (**31**), 11-dehydrosinulariolide (**32**), and sarcophytolide (**33**).

**Figure 11 marinedrugs-21-00451-f011:**
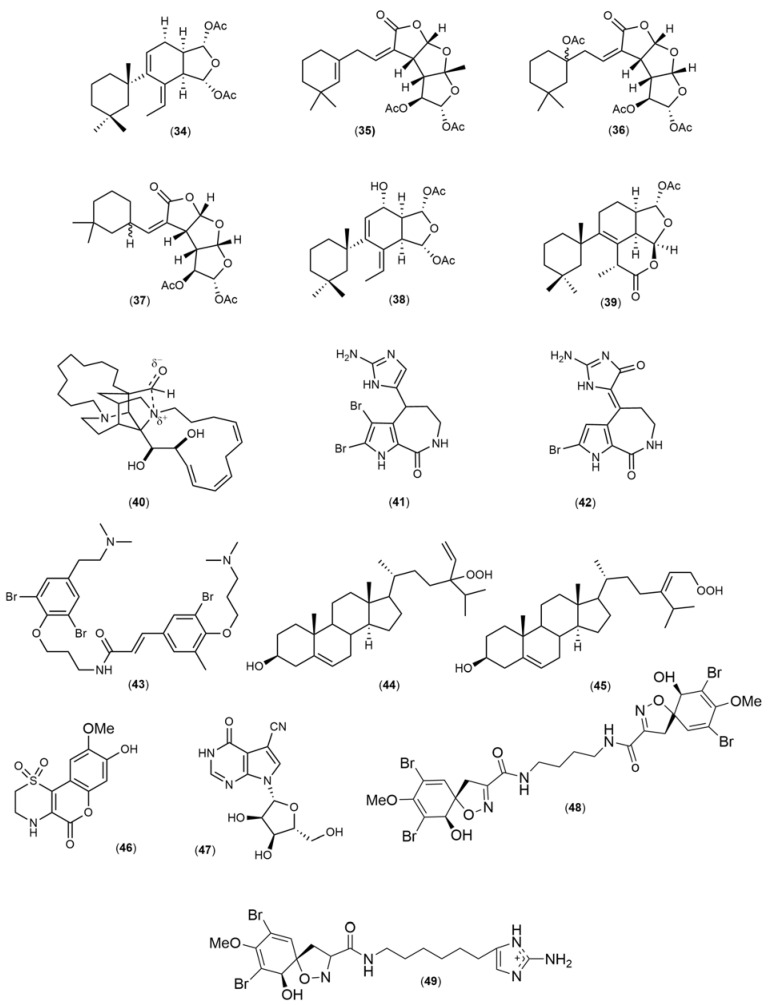
Chemical structures of marine natural products isolated from sponges: gracilins (A, H, K, J and L) (**34**–**38**), tetrahydroaplysulphurin-1 (**39**), sarain A (**40**), hymenin (**41**), hymenialdisine (**42**), psammaplysene A (**43**), 24-hydroperoxy-24-vinylcholesterol (**44**), 29-hydroperoxystigmasta-5,24(28)-dien-3-ol (**45**), iotrochotazine (**46**), jaspamycin (**47**), aerothionin (**48**), and aerophobin-2 (**49**).

**Figure 12 marinedrugs-21-00451-f012:**
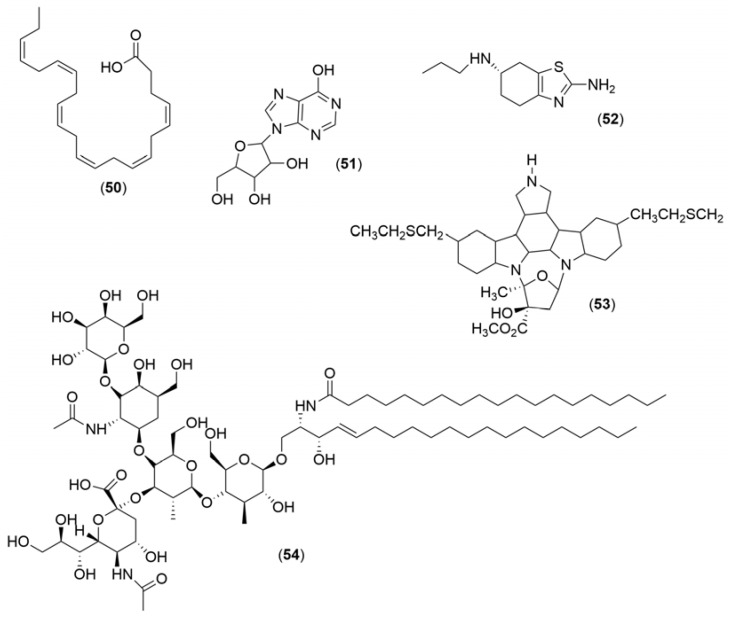
Chemical structures of marine-derived products used in clinical trials against Parkinson’s disease: docosahexaenoic acid (**50**), inosine (**51**), pramipexole (**52**), CEP-1347 (**53**), and ganglioside GM1 (**54**).

**Table 1 marinedrugs-21-00451-t001:** Neuroprotective activities of compounds isolated from marine organisms and their mode of action on in vitro and in vivo Parkinson’s disease models. Cross BBB: (−) cannot cross BBB or there are no available studies; (+) can cross BBB.

Species	Compound	Models	Effects/Mode of Action	Cross BBB	Reference
**Bacteria**					
*Streptomyces* sp.	NP7 (**1**)	In vitro	Inhibition of H_2_O_2_-induced neurotoxicity on neuronal-enriched and glial mesencephalic cultures; prevention of ERK and AKT phosphorylation induced by H_2_O_2_.	+	[67,68,69]
n.d.	Mannosylglycerate (**2**)	In vitro	Inhibition of α-synuclein aggregation on *Saccharomyces cerevisiae* cells.	−	[70]
*Streptomyces* sp.	Piloquinones A (**3**)	In vitro	Inhibition of monoamine oxidase (A and B) activity.	−	[71]
	Piloquinones B (**4**)			−	
**Fungi**					
*Eurotium rubrum* *Microsporum* sp. *Aspergillus* sp.	Neoechinulin A (**5**)	In vitro	Inhibition of MPP^+^/rotenone-induced neurotoxicity on PC12 cells; inhibition of mitochondrial Complex I dysfunction.	−	[72,73]
*Xylaria* sp.	Xyloketal B (**6**)	In vitro	Neuroprotective effects on PC12 cells against MPP^+^-induced neurotoxicity mediated by an increase of cellular viability, reduction of intracellular ROS accumulation, loss of mitochondrial membrane potential, and restoration of total GSH level.	+	[74,75]
*Aspergillus ochraceus* *Paecilomyces* sp.	Secalonic acid A (**7**)	In vitro	Neuroprotection of SH-SY5Y cells against MPP^+^-induced neurotoxicity mediated by an increase of cellular viability, down-regulation of Bax expression, suppression of Caspase-3 activation and inhibition of the phosphorylation of JNK and p38 MAPK; capacity to increase the number of dopaminergic neurons and upregulated striatal dopamine in 1-methyl-4-phenyl-1,2,3,6-tetrahydropyridine (MPTP)-induced Parkinson’s disease mice.	−	[76]
*Penicillium* sp.	6-Hydroxy-*N*-acetyl-β-oxotryptamine (**8**)	In vitro	Neuroprotective effects against 6-OHDA-induced neuronal cell death.	−	[77]
	3-Methylorsellinic acid (**9**)			−	
	8-Methoxy-3,5-dimethylisochroman-6-ol (**10**)			−	
*Aspergillus* sp.	Candidusin A (**11**)	In vitro		−	
	4”-Dehydroxycandidusin A (**12**)			−	
*Aspergillus flocculosus*	Diketopiperazine mactanamide (**13**)	In vitro	Neuroprotective effects against 6-OHDA-induced neuronal cell death.	−	[77]
*Aspergillus ochraceus* *Penicillium* sp.	Asperpendoline (**14**)	In vitro	Neuroprotective effect against oxidative stress in SH-SY5Y cells through reduction of ROS and augmented glutathione.	−	[78]
**Mollusks**					
*Sthenoteuthis oualaniensis*	Omega-3 fatty acids (Docosahexaenoic acid (**15**)/Eicosapentaenoic acid (**16**))	In vivo	DHA/EPA-PLs supplementation recovers brain DHA levels and exerts neuroprotective effects MPTP induced neurotoxicity on mice model. Protective effect on male C57BL/6J mice, on PD symptoms when MPTP-induced, increasing the number of dopaminergic neurons. Pretreatment with DHA could inhibit apoptosis via mitochondria-mediated pathway and MAPK pathway, and thus protecting dopaminergic neurons in MPTP-induced mice. Neuroprotective effect of DHA/EPA-PL on male C57BL/6J mice MPTP-induced, improving movement disorder, the oxidant–antioxidant status of the brain by reduced intracellular GSH levels and antioxidant enzyme activities, and increase intracellular lipid peroxidation; Reducing the phosphorylation of p38 and Jun N-terminal kinase.	+	[79,80,81]
*Doryteuthis singhalensis*	Astaxanthin (**17**)	In vitro	The astaxanthin treatment attenuated rotenone induced cytotoxicity, mitochondrial dysfunction, and oxidative stress in SKN- SH cells.	+	[82,83]
**Sea snails**					
*Conus textile*	α-Conotoxin (**18**)	In vivo	Regulation of DA release by blocking Nicotinic acetylcholine receptors α6/α3β2β3 nAChR expression on rat nAChRs model.	−	[84]
*Sepia pharaonis*	Sulfated chitosan	In vitro	Neuroprotective effect on SH-SY5Y cells against rotenone–induced neurotoxicity mediated by anti-apoptotic reduction of the intracellular ROS level, normalization of antioxidant enzymes, and mitigation of mitochondrial dysfunction and apoptosis.	+	[85,86]
*Neptunea arthritica cumingii*	YIAEDAER	In vivo	Neuroprotective effect through suppression of locomotor impairment, amelioration of DA neuronal degeneration, inhibition of vasculature and loss of cerebral vessels, suppression of α-syn levels, modulation of several autophagy-related genes, and modulation of oxidative stress in an MPTP-induced zebrafish.	−	[49]
**Sea cucumber**					
*Cucumaria frondosa*	Glucocerebrosides -1 (**19**)	In vitro	Neuritogenic effects on PC12 cells in a dose-dependent and structure-selective manner, increasing the ratio of neurite-bearing cells and expression of axonal (GAP-43) and synaptic (synaptophysin) proteins.	+	[87,88]
	Glucocerebrosides- 2 (**20**)			−	
	Glucocerebrosides- 3 (**21**)			−	
*Holothuria leucospilota*	Decanoic acid (**22**)	In vivo	Neuroprotective effect in 6-OHDA-induced *C. elegans* through attenuation of DAergic neurodegeneration, reduction of oxidative stress, and upregulation of transcriptional activity of DAF-16 targeted genes. Neuroprotective effect in an overexpressing α-synuclein-*C. elegans* model through reduction of α-synuclein aggregation, improved motor deficits, recovery of lipid deposition, and activation of mRNA expression of sod-3 and hsp16.2.	−	[89]
*Holothuria leucospilota*	Palmitic acid (**23**)	In vivo	Neuroprotective effect on *C. elegans* against 6-OHDA-induced neurotoxicity through restoration of dopaminergic neurons viability, improved dopamine-dependent behaviors, reduced oxidative stress, and prolonged lifespan; Neuroprotective effect on α-synuclein-based *C. elegans* model through improved locomotion, reduction of lipid accumulation, and extended lifespan.	+	[90,91]
*Cucumaria frondosa*	Eicosapentaenoic acid (**16**)	In vivo	Neuroprotective effect on male C57BL/6J mice improving MPTP-induced behavioral deficiency, suppressing oxidative stress, apoptosis and alleviating the loss of dopaminergic neurons via mitochondria-mediated pathway and mitogen-activated protein kinase pathway.	−	[92]
**Seaweeds**					
*Sargassum integerrimum* *Sargassum naozhouense* *Sargassum fusiforme*	Heteropolysaccharides	In vitro	Neuroprotective effect on MES 23.5 cells against 6-OHDA-induced neurotoxicity.	−	[93]
*Saccharina japonica*	Fucoidan	In vitro	Neuroprotective effect on MES 23.5 and SH-SY5Y cells against 6-OHDA-induced neurotoxicity.	+	[94,95]
*Laminaria japonica*	Fucoidan	In vitro/In vivo	Neuroprotective effect on dopaminergic cell line (MN9D) against MPP^+^-induced neurotoxicity and also on MPTP-induced animal model of Parkinsonism (C57/BL mice); Fucoidan attenuated MPTP-induced neurotoxicity on mice; Neuroprotective effect in MES23.5 cells against H_2_O_2-_induced neurotoxicity mediated by apoptosis, inhibition increasing Bcl-2/Bax ratio and decreasing the expression of Caspase-3 in the PI3K/Akt signaling pathway.	−	[96,97,98]
*Porphyra haitanensis*	Porphyran	In vitro	Neuroprotective effects in MES23.5 cells against 6-OHDA-induced neurotoxicity.	−	[99]
*Sargassum hemiphyllum*	Fucoidan	In vitro	Neuroprotective effects in SH-SY5Y cells against 6-OHDA-induced neurotoxicity decreasing cytochrome *C* release, Caspase-3; -8; -9 activity, protecting DNA fragmentation and phosphorylation of Akt.	−	[100]
n.d.	Fucoxanthin (**24**)	In vitro/In vivo	Neuroprotective effect in PC-12 cells against 6-OHDA-induced neurotoxicity reduction in mitochondrial membrane potential, suppressing ROS over-expression, and inhibiting activity of ERK/JNK-c-Jun system and expression of Caspase-3 protein; In the PD mice model, the long-term administration of fucoxanthin with L-dopa enhanced the motor ability.	+	[101]
*Ecklonia cava*	Dieckol (**25**)	In vitro	Neuroprotective effects in SH-SY5Y cells against rotenone-induced oxidative stress decreasing intracellular reactive oxygen species (ROS), cytochrome *C* release and retarding α-synuclein aggregation in α -synuclein overexpressing SH-SY5Y cells.	+	[102,103]
n.d.	Polymannosic acid (**26**)	In vivo	Neuroprotective effects in an in vivo model of SNpc, improving motor functions and preventing dopaminergic neuronal loss; Ability to improve striatal neurotransmitters of 5-HT and 5-HIAA levels.	−	[104]
	Polyguluronic acid (**27**)	In vivo	Neuroprotective effects in SNpc model of PD mice, increasing TH expressions in the SNpc. Ability to improve striatal neurotransmitters of 5-HT.	−	
*Bifurcaria bifurcata*	Eleganolone (**28**)	In vitro	Neuroprotective effects on SH-SY5Y cells against 6-OHDA induced neurotoxicity decreasing ROS production, Caspase-3 activity, protecting mitochondrial membrane potential, increasing Catalase activity, ATP levels, and inhibited blocking translocation of NF-kB factor; Eleganolone also attenuated LPS -induced inflammation on RAW 264.7 macrophages cells, decreasing NO levels and TNF-α and IL-6 interleukins levels.	−	[105]
*Turbinaria decurrens*	Fucoidan	In vivo	Neuroprotective effect on in vivo rat model against MPTP-induced neurotoxicity.	−	[97]
-	κ-Carrageenan	In vitro	Protection of microglial cells through the reduction of TNF-α and arginase levels.	−	[106]
*Hypnea musciformis*	κ-Carrageenan	In vitro	Neuroprotective effect on SH- SY5Y cells against 6-OHDA-induced neurotoxicity protecting mitochondrial membrane potential and decreasing Caspase-3 activity.	−	[107]
*Gracilaria cornea*	Agaran	In vivo	Neuroprotective effect on in vivo rat model against 6-OHDA-induced neurotoxicity, reducing the oxidative/nitroactive stress and alterations in the MAO contents.	−	[108]
*Codium tomentosum*	Loliolide (**29**)	In vitro	Neuroprotective effects on SH-SY5Y cells against 6-OHDA induced neurotoxicity, decreasing ROS production, Caspase-3 activity, protecting mitochondrial membrane potential, DNA fragmentation and inhibited. blocking translocation NF-kB factor; Attenuates LPS-induced inflammation on RAW 264.7 macrophages cells, decreasing NO levels and TNF-α and IL-6 interleukins levels.	+	[109,110]
*Fucus vesiculosus*	Fucoidan	In vivo	Neuroprotective effect through the improvement of mitochondrial dysfunction, prevention of neuronal apoptosis, reduction of dopaminergic neuronal loss, and improvement of motor deficits in an (MPTP)-induced PD mouse model.	−	[111]
**Soft coral**					
*Cladiella australis*	Austrasulfone (**30**)	In vitro	Neuroprotective effect on SH-SY5Y cells against 6-OHDA-induced neurotoxicity.	−	[112]
	1-Tosylpentan-3-one (**31**)	In vitro	Neuroprotective effect on SH-SY5Y cells against 6-OHDA-induced neurotoxicity mediated by activation of both p38 MAPK, and a decrease of Caspase-3 and nuclear factor erythroid 2-related factor 2 (Nrf2) expression.	−	[113]
*Sinularia flexibilis*	11-Dehydrosinulariolide (**32**)	In vitro	Neuroprotective effect on SH-SY5Y cells against 6-OHDA-induced neurotoxicity mediated by anti-apoptotic and anti-inflammatory actions via PI3K signaling pathway.	−	[114]
*Sarcophyton glaucum*	Sarcophytolide (**33**)	In vitro	Neuroprotective effect on primary cortical cells from rat embryos against glutamate–induced neurotoxicity mediating an increase of the proto-oncogenebcl-2 (BCl-2) expression.	−	[115]
**Sponges**					
*Spongionella gracilis*	Gracilin A (**34**)	In vitro	Inhibition of H_2_O_2_-induced neurotoxicity on SH-SY5Y cells; Activation of Nrf2/ARE pathways.	+	[116,16]
	Gracilin H (**35**)			−	
	Gracilin J (**36**)			−	
	Gracilin K (**37**)			−	
	Gracilin L (**38**)			−	
	Tetrahydroaplysuphurin-1 (**39**)			−	
*Haliclona (Rhizoniera) sarai*	Sarain A (**40**)	In vitro	Inhibition of H_2_O_2_-induced neurotoxicity on SH-SY5Y cells; Block the mPTP and enhance the Nrf2 pathway.	−	[16]
n.d.	Hymenin (**41**)	In vitro	Inhibition of H_2_O_2_-induced neurotoxicity in primary cortical neurons; Decrease of ROS production; Increase of GSH levels; Induces the translocation of the Nrf2 factor to the nucleus.	+	[117]
	Hymenialdisine (**42**)			−	
*Psammaplysilla* sp.	Psammaplysene A (**43**)	In vitro/In vivo	Neuroprotective effects in HEK293 cells and Drosophila in vivo model mediated by modifying processes dependent on heterogeneous nuclear ribonucleoprotein (HNRNPK) protein that controls biological aspects of RNA.	−	[118]
*Xestospongia testudinaria*	24-Hydroperoxy-24-vinylcholesterol (**44**)	In vitro	Inhibition of NF-kB factor transcription activation.	+	[119]
	29-Hydroperoxystigmasta-5.24(28)-dien-3-ol (**45**)	In vitro	Inhibition of H_2_O_2_-induced neurotoxicity on SH-SY5Y cells.	−	
*Iotrochota* sp.	Iotrochotazine A (**46**)	In vivo	Acts on the early endosome and lysosome markers in idiopathic PD patients.	−	[120]
*Jaspis splendens*	Jaspamycin (**47**)	In vivo	Increase of lysosomal staining and decrease of the number of early endosomes associated with EEA-1 on a hONS cellular model of PD.	−	[121]
*Aplysinella* sp.	Aerothionin (**48**)	In vitro	Inhibits α-syn aggregation in a ThT aggregation assay.	−	[122]
	Aerophobin-2 (**49**)	In vivo	Inhibited α-syn aggregation in primary dopaminergic mouse neurons.	−	
**Starfish**					
*Asterias rollestoni*	Glucan	In vitro	Inhibition of 6-OHDA-induced neurotoxicity on MES23.5 cells.	+	[123]
**Several microorganisms (fish and crustaceans)**					
−	Astaxanthin (**18**)	In vitro/in vivo	The astaxanthin treatment the inhibited cell death and apoptosis promotion induced by 1-methyl-4-phenylpyridinium (MPP+) in SH-SY5Y cells via inhibiting endoplasmic reticulum (ER) stress and reversed the MPP+ caused dysregulation of miR-7 and SNCA expression. The astaxanthin treatment inhibited cell death induced by MPP+ in PC-12 cells, and oxidative stress, via the SP1/NR1 signaling pathway, through decreased expression of Sp1 and NR1 mRNA.	+	[124,125]

**Table 2 marinedrugs-21-00451-t002:** Marine-derived products that reached the clinical trials stage for Parkinson´s disease treatment (Sources: ClinicalTrials.gov, accessed on 11 July 2023).

Name	Source	Clinical Trial.gov Identifier	Study Star	Study Completion	Phase
Docosahexaenoic acid (**50**)	Fish and algae	NCT01563913	October 2012	June 2016	1
Inosine (**51**)	Sponge	NCT02642393	June 2016	June 2019	3
Pramipexole (**52**)	Marine yeasts	NCT00197374	August 2012	September 2017	4
CEP-1347 (**53**)	*Nocardiopsis* sp. (Marine bacteria)	NCT00040404	March 2002	August 2005	2/3
CEP-1347 (**53**)		NCT00404170	November 2006	July 2014	2
GM1 Ganglioside (**54**)	*Pseudomonas* sp.	NCT00037830	November 1999	June 2010	2

## Data Availability

Not applicable.

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
