# Peer review of "Marine-Derived Components: Can They Be a Potential Therapeutic Approach to Parkinson’s Disease?"

_marinedrugs, 2023, doi:10.3390/md21080451_

Round 1
Reviewer 1 Report
- The attention payed to the possible use of marine-derived compounds to neurological disorders is appreciable. This review is well structured. Some points should be addressed:
- the title refers to macroalgae, the manuscript examines and describes several marine organisms. Why? if the authors consider macroalgae as much relevant in comparison to other it should clearly emerge; on the contrary the title should be modified
- the authors consider the isolate compounds. It should be very interesting also to think to the possibility to use for therapeutic purposes an extract of the marine organism. I mean as we currently do with botanicals where the phytocomplex (or part of that) is used as mix of several molecules. It allows in several case additive or synergic effects. This point should be discussed. For sure sustainability problems should be considered but they could be overcome by dedicated culture of organisms.
- the possible role of marine-derived compounds against neurodegenerative disorders was partly discussed in previous papers (e.g.DOI: 10.2174/1871527321666220511205231), please check that these articles are correctly cited in your work
- if possible improve the graphical aspect of the cartoon related to Parkinson diseases mechanisms
please revise
Author Response
The attention payed to the possible use of marine-derived compounds to neurological disorders is appreciable. This review is well structured. Some points should be addressed:
- the title refers to macroalgae, and the manuscript examines and describes several marine organisms. Why? if the authors consider macroalgae as much relevant in comparison to other it should clearly emerge; on the contrary the title should be modified.
R: We appreciate and agree with the reviewer's comment. The title has been changed. Changes are highlighted in yellow in the mansucript (Line 2 - 3)
- the authors consider the isolate compounds. It should be very interesting also to think to the possibility to use for therapeutic purposes an extract of the marine organism. I mean as we currently do with botanicals where the phytocomplex (or part of that) is used as mix of several molecules. It allows in several case additive or synergic effects. This point should be discussed. For sure sustainability problems should be considered but they could be overcome by dedicated culture of organisms.
R: We appreciate the reviewer's comment. Our focus was centered in the potential of isolated molecules, through which it is possible to understand the mechanism of action behind of the activity and could inspire the development of new analogues with therapeutic potential. In opposite, the use of extracts, if not produced in controlled conditions, could be influenced by some factors like seasonality compromising the activities observed. In the case of botanicals, there is a previous empiric knowledge about of these mixtures that have been ancient times
- the possible role of marine-derived compounds against neurodegenerative disorders was partly discussed in previous papers (e.g.DOI: 10.2174/1871527321666220511205231), please check that these articles are correctly cited in your work.
R: Thanks for sharing this article was taken into account, however, it is not available for consultation, and it is not possible to verify whether all the compounds described in this manuscript were also reported in the article referred to by the reviewer.
- if possible, improve the graphical aspect of the cartoon related to Parkinson diseases mechanisms
R: The figure 2 was improved as suggested by the reviewer (167 – 189).

Reviewer 2 Report

The article should be read by an English speaking colleague are minor corrections made.
Author Response
The manuscript describes about 60 molecular structures extracted from various marine species that could be possible candidates for addressing PD. The validity for such intervention is based on effects on one or more PD targets (stated as oxidative stress, mitochondrial dysfunction, alpha-synuclein aggregation, and inflammatory pathways). The authors have provided extensive tables listing useful compounds together with the mode of action on specific targets associated with PD.
They need to clearly identify which of the studies used in-vitro models and which relate to animal models of PD.
R: The Table I as improved as suggested by the reviewer. The information about of each type study, in vitro and in vivo, that was carried out as added (line 233).
At the present, for the compound to be taken seriously for intervention in PD, it should be able to cross the blood-brain barrier. Obviously, the use of nano-particle delivery systems is a futuristic consideration. How many of the listed compounds can cross the blood-brain barrier?
To bring the manuscript up-to-date, the authors should include a list of human clinical trials using these marine drugs as potential vehicles for addressing PD. A description of the outcome of these trials together with futuristic possibilities should be discussed.
R: Thank you very much for your comment. The information about the ability of the marine compounds described in table I to cross the blood-brain barrier was added in the table (column "cross BBB").
R: Marine compounds that have already entered clinical trials was also reviewed and a new topic “Marine - derived compounds for clinical trial of Parkinson Disease” was added to the manuscript and accompanied by table II and figure 11. Changes are between the line 591 – 640.
Minor points:
Page 1. Algae derived components are minor in the list of 60 or so structures given. Therefore the manuscript title should be broader such as…..marine derived components.
R: The title was improved. Line 2-3.
Page 3. Figure 2 is not relevant and should be removed.
R: Figure 2 was removed.
Pages 6-10: Mode of action section is well described. Should indicate which of these studies are in-vitro and which refer to animal models.
R: The Table I as improved as suggested by the reviewer. The information about of each type study, in vitro and in vivo, that was carried out as added (line 233).
Overall, the English requires a little attention.
R: We thank the referee pertinent suggestions and corrections. The manuscript was revised, and the changes were highlighted using the tracking mode.

Round 2
Reviewer 2 Report
Some clarification is required as to the meaning of (+) and (-) in Table 1 associated with the ability of the compound to cross the BBB. How can you get (+) from an in-vitro study? Or does the (+) relate to some other in-vivo study that has not been cited?
Does (-) mean that the drug does not cross the BBB or that it has not been evaluated yet?
In Table 2, compound Inosine (structure 46) was used in the clinical trial. In Table 1, compound 46 is called Aerophobin-2. Are these two compounds the same? Again, in Table 1, this compound does not cross the BBB and yet it is used in the clinical trial. Please clarify.
What you mean by (+) and (-) can be added to the Legend to Table 1.
Some very minor corrections required. It would be best if an English speaking colleague read through the paper and made the minor corrections.
Author Response
Some clarification is required as to the meaning of (+) and (-) in Table 1 associated with the ability of the compound to cross the BBB. How can you get (+) from an in-vitro study? Or does the (+) relate to some other in-vivo study that has not been cited?
Does (-) mean that the drug does not cross the BBB or that it has not been evaluated yet?
R: (-) means that the compound does not have the ability to cross the BBB or there are no studies. (+) means that the compound could cross the BBB.
The meaning of these two symbols (-) and (+) can be found in the table caption on line 237
In Table 2, the compound Inosine (structure 46) was used in the clinical trial. In Table 1, compound 46 is called Aerophobin-2. Are these two compounds the same? Again, in Table 1, this compound does not cross the BBB and yet it is used in the clinical trial. Please clarify.
What you mean by (+) and (-) can be added to the Legend to Table 1.
R: There is an error in table 2, compound 46 is not the same as compound 46 found in table 1. The continuation of the numbering is wrong. In this way, the numbering of the compounds was rectified in table 2 and figure 2.
Comments on the Quality of English Language
Some very minor corrections are required. It would be best if an English-speaking colleague read through the paper and made the minor corrections.
R: As suggested, English has been rectified again in the manuscript.
